# The Effect of Beetroot Ingestion on High-Intensity Interval Training: A Systematic Review and Meta-Analysis

**DOI:** 10.3390/nu13113674

**Published:** 2021-10-20

**Authors:** Tak Hiong Wong, Alexiaa Sim, Stephen F. Burns

**Affiliations:** Physical Education and Sports Science, National Institute of Education, Nanyang Technological University, 1 Nanyang Walk, Singapore 637616, Singapore; nie21.wth@e.ntu.edu.sg (T.H.W.); nie184704@e.ntu.edu.sg (A.S.)

**Keywords:** nitrate, nitrite, nitric oxide, beetroot, high-intensity interval training, sprint interval training

## Abstract

Dietary nitrate supplementation has shown promising ergogenic effects on endurance exercise. However, at present there is no systematic analysis evaluating the effects of acute or chronic nitrate supplementation on performance measures during high-intensity interval training (HIIT) and sprint interval training (SIT). The main aim of this systematic review and meta-analysis was to evaluate the evidence for supplementation of dietary beetroot—a common source of nitrate—to improve peak and mean power output during HIIT and SIT. A systematic literature search was carried out following PRISMA guidelines and the PICOS framework within the following databases: PubMed, ProQuest, ScienceDirect, and SPORTDiscus. Search terms used were: ((nitrate OR nitrite OR beetroot) AND (HIIT or high intensity or sprint interval or SIT) AND (performance)). A total of 17 studies were included and reviewed independently. Seven studies applied an acute supplementation strategy and ten studies applied chronic supplementation. The standardised mean difference for mean power output showed an overall trivial, non-significant effect in favour of placebo (Hedges’ g = −0.05, 95% CI −0.32 to 0.21, Z = 0.39, *p* = 0.69). The standardised mean difference for peak power output showed a trivial, non-significant effect in favour of the beetroot juice intervention (Hedges’ g = 0.08, 95% CI −0.14 to 0.30, Z = 0.72, *p* = 0.47). The present meta-analysis showed trivial statistical heterogeneity in power output, but the variation in the exercise protocols, nitrate dosage, type of beetroot products, supplementation strategy, and duration among studies restricted a firm conclusion of the effect of beetroot supplementation on HIIT performance. Our findings suggest that beetroot supplementation offers no significant improvement to peak or mean power output during HIIT or SIT. Future research could further examine the ergogenic potential by optimising the beetroot supplementation strategy in terms of dosage, timing, and type of beetroot product. The potential combined effect of other ingredients in the beetroot products should not be undermined. Finally, a chronic supplementation protocol with a higher beetroot dosage (>12.9 mmol/day for 6 days) is recommended for future HIIT and SIT study.

## 1. Introduction

Nitric oxide (NO) is a signalling molecule that contributes to numerous physiological functions, including mitochondrial respiration and biogenesis, vasodilation, muscle glucose uptake, angiogenesis, and sarcoplasmic reticulum calcium handling [1,2]. NO is produced from the conversion of the amino acid L-arginine to L-citrulline in the presence of oxygen. If not used locally it is oxidised to nitrate. Importantly, the nitrate produced from the endogenous oxidation of NO can be reduced to nitrite and then back to NO in a reaction which is facilitated by acidic and low-oxygen environments. In addition, exogenous inorganic nitrate obtained from the diet can act as another source of substrate which can also eventually be reduced to NO via this pathway [3].

Beetroot (of the family Chenopodiaceae) is a good source of dietary nitrate with an average value of 1446 mg per kg of fresh weight (range of 214–3556 mg/kg) [4]. A recent International Olympic Committee (IOC) consensus statement [5] supported the supplementation of high nitrate containing foods or juices for exercise performance in several areas [6,7,8,9,10]. These include extending time to exhaustion in endurance exercise, improving time trials <40 min in duration, enhanced type II muscle fibre function during high intensity exercise, and improving intermittent team sport activities [5,6,7,8,9,10]. To date, most studies examining exercise performance have been conducted in trained athletes and have used beetroot juice as the supplemental source of dietary nitrate [6,11,12,13,14,15], although spinach, arugula, and celery are other sources. Rather than dietary nitrate sources, some studies have directly supplemented sodium or potassium nitrate—as this represents a one rather than two-step reduction to NO with a higher conversion efficiency—but these failed to show ergogenic effects on exercise performance [12,16,17,18,19,20]. It was hypothesised that additional compounds contained in the beetroot juice supplementation products play certain roles in nitrate reduction and metabolism [20]. Beetroot juice contains a high amount of polyphenols and ascorbic acid, which ease NO production in the gut [21] and have a greater potency in reducing blood pressure compared to sodium or potassium nitrate [22].

Recommended dietary nitrate intake—from products such as beetroot juice supplementation—for sports performance ranges from 0.1 to 0.2 mmol/kg body weight depending on the supplementation strategy and timing, with plasma nitrate and nitrite elevated one- to five-fold compared to placebo supplementation [12,23]. A study of the chemical generation of NO in enterosalivary circulation suggested that the amount of NO generated in the mouth and stomach and the salivary nitrate concentration rises considerably and remains elevated for several hours after dietary nitrate ingestion. Conversion of nitrite to NO depends on the number and activity of nitrate-reducing bacteria on the tongue [24]. In the stomach, nitrite is further decomposed to NO under acidic conditions. The remaining nitrate and nitrite are absorbed from the intestine into the blood circulation and later can become bioactive NO in the muscles and blood under hypoxic conditions [15]. It has also been suggested that the remaining nitrate and nitrite can be preserved and stored in the blood and muscle tissues as a “NO reservoir”, which can be drawn upon for ongoing vascular and metabolic processes, and which potentially contribute to optimal physical and cognitive functions [3]. In addition, the rate of generation of NO is enhanced by ascorbic acid [24]. Ascorbic acid and polyphenols from beetroot juice can reduce nitrite to NO in the stomach [25,26] and another study described that a mixture of sodium nitrite and ascorbic acid would rapidly release NO and other nitrogen oxides [27]. In the presence of ascorbic acid and polyphenols, the nitrate reduction to NO is greatly enhanced with less generation of other nitrogen oxides, such as dinitrogen trioxide ($N_{2}O_{3}$) and nitrogen dioxide (${\mathrm{NO}}_{2}^{-}$) [27].

A variety of enzymes and proteins, including deoxyhaemoglobin, catalyse nitrite reduction in the blood, and other tissues. As stated, NO production is facilitated in low pH and low oxygen environments similar to the hypoxic conditions which occur in skeletal muscle during high-intensity exercise [1]. It is suggested that NO, being a major vasodilator, induces physiological responses by increasing blood flow to muscles and influences oxygen utilisation during skeletal muscle contraction [15]. Moreover, NO is thought to improve glucose uptake and mitochondrial efficiency in muscle [28], and supplement muscle contraction and relaxation processes [29]. Indeed, animal studies have shown nitrate enhanced exercise performance through effects on contractile function and blood flow in type II muscle fibres [3,29,30].

In humans, dietary beetroot supplementation can improve submaximal endurance exercise performance. For example, in healthy trained adults, ingestion of 0.1 mmol/kg nitrate was associated with reduced oxygen cost during submaximal cycling of 5.4%, and increased energy efficiency of 7.1% [12]. Another study concluded the aftereffect of dietary nitrate consumption (6.2 mmol nitrate supplement for six days) on low, medium, and high intensity running/exercise and noted that whilst oxygen demand was reduced by 20, 7.1, and 7.2%, respectively, time to fatigue was increased 15% during high intensity running (75% of the difference between the running speed at the Gas Exchange Threshold (GET) and VO_2max_ + the running speed at the GET) [23]. A higher dose of nitrate of 11.2 mmol reduced muscle fractional oxygen extraction—estimated from near-infrared spectroscopy—by 19%, whilst improving time to exhaustion during a severe cycling task by 15.7% relative to placebo [10]. Given that exercise performance with nitrate supplementation from beetroot could be enhanced by increasing oxygen and substrate delivery to the working skeletal muscle [31] it is important to extend these observations beyond endurance exercise. Indeed if the conversion of nitrate to NO is enhanced under conditions of hypoxia and raised acidity, then the effects of beetroot on performance may potentially be greatest during sprint exercise where it could accelerate recovery via improved oxygen delivery between sprints [31].

From this perspective, several studies have focused on the effects of nitrate supplementation during high-intensity interval training (HIIT) or sprint interval training (SIT) related protocols [8,32,33,34,35,36]. To date, the evidence for an ergogenic effect of nitrate on HIIT and SIT remains equivocal. One systematic review including nine articles indicated that acute and chronic ingestion of beetroot juice might improve performance during intermittent, high-intensity efforts with short rest periods [37]. It was suggested that power output may be improved with consumption of beetroot juice via several mechanisms. These include better release and re-uptake of calcium at the sarcoplasmic reticulum which assists with a faster muscle shortening velocity, faster phosphocreatine resynthesis between high-intensity efforts, and/or a reduced build-up of metabolites associated with muscular fatigue [37]. However, not all studies have demonstrated performance improvements during high-intensity efforts after beetroot juice ingestion. For example, Pawlak-Chaouch and colleagues found no significant improvement in mean power, total work completed, and total repetitions with the consumption of beetroot juice in 11 elite endurance athletes undertaking 15 s cycling bouts at 170% of maximal aerobic power [36]. Similarly, Muggeridge et al. found no effect of nitrate-rich concentrated beetroot juice on peak power in five 10 s kayak sprints in trained kayakers [38]. As already noted, a recent IOC consensus statement supports the use of dietary nitrate [5] (including beetroot juice) for performance in higher intensity exercise, including intermittent team sport activities [5,6,7,8,9,10]. Nonetheless, the effect of beetroot juice on performance measures (e.g., power output) during repeated sprint type activity such as HIIT or SIT is unclear. It is important to systematically review and evaluate the effects of beetroot on performance in high intensity exercise so that coaches and athletes involved in sports which employ this type of training or activity have greater assurance of its ergogenic benefits. In addition, such a review will provide valuable evidence on the prescription of beetroot in terms of type, dosage, and for which types of athlete it may be most valuable. 

While studies on dietary nitrate via beetroot supplementation and its effects on HIIT and SIT are available, the conclusions from these studies remain inconclusive and unclear. Thus, the main aim of the present systematic review and meta-analysis was to examine the acute and chronic ergogenic effect(s) of dietary nitrate from beetroot juice on mean and peak power output during repeated work bouts of HIIT and SIT performance. A secondary aim was to examine the acute and chronic ergogenic effects of dietary nitrate from beetroot juice on other markers of performance (e.g., distance covered) during HIIT or SIT.

## 2. Materials and Methods

This systematic review was conducted according to PRISMA (Preferred Reporting Items for Systematic Reviews and Meta-Analyses) guidelines (Appendix A) [39,40] and the PICOS (population, intervention, comparison, outcome, study design) framework [41]. Covidence (Covidence systematic review software, Veritas Health Innovation, Melbourne, Australia) was used as the tool for title/abstract and full-text screening. 

### 2.1. Search Strategy

A systematic literature search was carried out from four databases: PubMed, ProQuest, ScienceDirect and SPORTDiscus. The search terms used were: ((nitrate OR nitrite OR beetroot) AND (HIIT or high intensity or sprint interval or SIT) AND (performance)). The period of the search was from 1 January 2010 to 31 March 2021. The search yields were imported to Covidence for screening. Three additional studies [42,43,44] which reported power output as part of the performance outcomes through reference list searches [15,45] were included. After the elimination of duplicates, a total of 138 studies were identified and reviewed independently. Figure 1 summarises the identification of studies using the PRISMA Flow Diagram. 

### 2.2. Inclusion and Exclusion Criteria

Inclusion criteria for all relevant studies were defined based on the PICOS model (Population: Active adults 18–45 years old; Intervention: Beetroot supplementation; Comparison: Same conditions with placebo or control group; Outcome: Exercise performance measure; Study type: Randomised crossover (repeated measures) or Independent (parallel) group designs. The following inclusion criteria were applied: (1) full article; (2) beetroot and placebo/control intervention; (3) precise information on dosage and ingestion timing; (4) assessed and reported HIIT or SIT performance measures; (5) employed a randomised crossover (repeated measures) or parallel group designs; (6) healthy active adults 18–45 years old; (7) article published in English. No filters were applied in respect of athletic level, gender, or ethnicity.

Two researchers screened the title, abstract, and full paper independently for eligibility (THW and AS). Differences in opinion and included/excluded papers were resolved through discussion and consensus with a third reviewer (SFB). After the elimination of duplicates and screening of inclusion criteria, a total of 17 studies were identified for review and data extraction.

### 2.3. Data Extraction and Analysis

Data extraction was conducted manually using a standardised form (Microsoft Excel, 2008) by one researcher (THW) and checked by a second researcher (AS). Disagreement was resolved by discussion and consensus with input from the third researcher (SFB). Specific information extracted included authors, year, sample size, sex, age, exercise level, dosage, supplementation timing, supplement source, nitrate/nitrite concentration, study design, exercise protocol and primary outcome. Studies were grouped later by the primary outcome for meta-analysis. 

### 2.4. Quality Assessment

Study quality was assessed using the Physiotherapy Evidence Database (PEDro) scale, which provides a reliable assessment of internal validity [46]. Each eligible article was independently assessed by two reviewers (THW and AS) using the 11-item checklist to obtain a maximum score of 10 (item 1 on eligibility criteria does not contribute to the total score). Differences were resolved through discussion and consensus. Secondly, the risk of bias was checked using the Revised Cochrane risk-of-bias tool for randomised trials (RoB 2) [47]. The RoB 2 tool provides a framework to assess the risk of bias in study findings within five domains: (1) bias arising from the randomisation process; (2) bias due to deviations from intended interventions; (3) bias due to missing outcome data; (4) bias in the measurement of the outcome, and (5) bias in the selection of the reported result. Methodological quality and risk of bias were assessed by two researchers independently (THW and AS). Differences were resolved through discussion and consensus and clarified with a third Reviewer (SFB) if necessary.

### 2.5. Statistical Analysis

Participants and performance data are reported as mean and standard deviation. The level of agreement between researchers on study quality was evaluated using Cohen’s kappa statistic, and the relation between percentage increase of plasma nitrite and nitrate dosage was evaluated using Pearson correlation test (IBM SPSS Statistics for Windows, Version 26.0. Armonk, NY, USA: IBM Corp). Meta-analysis was performed using Review Manager (RevMan) version 5.4 (The Cochrane Collaboration, 2020). A random-effects model was applied to compute the standardised mean difference between intervention and placebo, calculated as Hedges’ g [48,49]. The overall effect (95% Confidence Interval (CI)) and I^2^ values (percentage of total variation among studies) were calculated by RevMan. Effect sizes are described as trivial (<0.2), small (<0.5), moderate (<0.8), and large (>0.8) [48]. The I^2^ values were guided as follows: (1) might not be important (0–40%); (2) may represent moderate heterogeneity (30–60%); (3) may represent substantial heterogeneity (50–90%), and (4) considerable heterogeneity (75–100%) [48].

## 3. Results

### 3.1. Internal Validity and Risk of Bias

The mean PEDro score was 7.4 (0.7) out of 10 (Appendix B). One study employed a counter-balanced, repeated measures design but did not clearly report how the randomisation of participants was achieved [35]. Two studies using 4-week chronic supplementation with beetroot juice used independent groups matched at baseline on physical characteristics, and physiological and performance variables of interest, but were also unclear how matched individuals were randomised into their groups [32,50]. Thirteen studies employed a double-blinded design, three studies a single-blinded design [36,42,44], whilst one study was not blinded as the researchers were unable to blind the participants to the distinctive taste of the drinks used [38]. However, although blinding was reported in most studies, none clearly explained how the process of blinding was achieved or the effectiveness. The level of agreement between reviewers was k = 0.903 (Kappa value), which can be interpreted as almost perfect agreement [51]. There was no disagreement between the reviewers in classifying studies using the RoB 2. Fourteen articles were considered “low risk”, and three articles were considered as “some concerns” in relation to the randomisation process. No studies were considered at “high risk” of bias (Appendix C).

### 3.2. Participant and Study Characteristics

A summary of the participants’ characteristics, study design, exercise protocol, and the primary outcome of the 17 studies included in this systematic review is provided in Table 1. The 17 included studies included 319 participants–19.4% (*n* = 62) females, 80.6% (*n* = 257) males. The largest sample size was 52 [33], and the smallest was 7 [52]. The mean age range of participants was from 20.7 ± 1.3 to 31.0 ± 15.0 years. The participants consisted of healthy adults (*n* = 7), recreational exercisers (*n* = 156), competitive/trained individuals (*n* = 147) and elite athletes (*n* = 9). 

All studies included placebo and beetroot (nitrate) intervention groups. Seven studies applied acute supplementation of beetroot juice on markers of performance whilst ten studies used a chronic supplementation strategy (beetroot juice provided on >1 day).

### 3.3. HIIT Protocols

Repeated all-out sprints of different timing ranging between 3 and 30 s (3, 4, 6, 10, 15, 20, and 30 s), interspersed with a mixture of passive, active, and mixed recovery between sprints ranging in duration from 22 s to 4 min were used in 14 studies [8,32,33,34,35,36,38,42,43,44,50,52,53,54]. The Yo-Yo Intermittent Recovery Test Level 1(Yo-Yo IR1) protocol was used in three studies [7,22,55]. The Yo-Yo IR1 evaluates an individual’s ability to perform repeated intense, intermittent exercise. It was designed to cause maximal activation of the aerobic system [56]. Briefly, it involves two 20 m shuttle runs at increasing speed interspersed with 10 s of active recovery between runs. The test is maintained until an individual is unable to match the speed and is therefore maximal in nature. 

Power output measures were the primary outcomes extracted followed by performance measures (distance covered, time trial, or time to exhaustion after HIIT or SIT). Six studies showed improved power output and performance after beetroot juice supplementation, while eleven studies found no significant difference when compared with placebo (Table 1).

### 3.4. Nitrate Supplementation

Information on the supplementation strategy, source of beetroot, and increases in nitrate/nitrite are provided in Table 2. Dosage of nitrate provided ranged from 4.84 mmol to 12.9 mmol per serving for acute studies and from 5.2 mmol per day to 29.0 mmol per 36 h for chronic intervention studies. Total nitrate intake in chronic studies ranged from 15.6 mmol provided over three days to 364.0 mmol over 28 days. Most studies gave the last dose of beetroot juice 2 to 3 h before the exercise intervention, except for one acute study which provided a top-up dose 1.5 h before exercise [22]. Fourteen studies provided the beetroot/nitrate drink from Beet It (James White Drink Ltd., Ipswich, UK), one study produced the beetroot juice in-house [54], and two studies sourced the beetroot juice from Pajottenlander TM, Belgium [36,42].

We examined the relationship between percentage of nitrite increase in response to the total dosage of nitrate received throughout the trial and the dosage per day of nitrate with chronic supplementation (Figure 2). A Pearson correlation test showed that the percentage increase of plasma nitrite was significantly related to total nitrate dosage throughout the trial (*r* = 0.878, *p* = 0.004, *N* = 8), but the relationship to dosage per day did not reach significance (*r* = 0.655, *p* = 0.078, *N* = 8).

### 3.5. Meta-Analyses

Ten studies reported peak power output, and eight studies reported mean power output outcomes. One study investigated three separate HIIT protocols [8]. These three protocols were included in the meta-analyses. Three studies reported distance covered in the Yo-Yo IR1 test [7,22,55] and were analysed separately based on their primary outcome measure.

Figure 3 displays the forest plot comparing the effect of acute and chronic beetroot supplementation on mean power output. The standardised mean difference for mean power output showed an overall trivial, non-significant effect in favour of placebo (Hedges’ g = −0.05, 95% CI −0.32 to 0.21, Z = 0.39, *p* = 0.69). Random effects analysis displayed trivial heterogeneity among studies (I^2^ = 0%; *p* = 0.82). Five analysed studies (two using acute supplementation and three employing chronic supplementation) presented positive effects on mean power output after beetroot intervention with the greatest improvement reported by Bernardi and colleagues [54]. Three studies reporting on four different sprint conditions showed negative effects of beetroot on mean power output [8,34,35] and one study favoured no difference between beetroot juice and placebo [44]. Subgroup analysis on the effect of acute beetroot supplementation on mean power output revealed that there was a trivial, non-significant effect in favour of placebo (Hedges’ g = −0.18, 95% CI −0.66 to 0.31, Z = 0.72, *p* = 0.47). The subgroup analysis of chronic supplementation showed a trivial, non-significant effect in favour of beetroot juice intervention (Hedges’ g = 0.04, 95% CI −0.31 to 0.40, Z = 0.23, *p* = 0.82). There was no heterogeneity between subgroups (I^2^ = 0%, *p* = 0.47).

Pooling data from eleven studies, the standardised mean difference for peak power output showed an overall trivial, non-significant effect in favour of the beetroot juice intervention (Hedges’ g = 0.08, 95% CI −0.14 to 0.30, Z = 0.72, *p* = 0.47) (Figure 4). Random effects analysis showed trivial heterogeneity among studies (I^2^ = 0%; *p* = 0.99). Eight studies presented positive effects on peak power output after beetroot intervention with the greatest improvement reported by Muggeridge and colleagues (2013). Three studies showed negative effects of beetroot on peak power output [8,34,35]. Subgroup analysis on the effect of acute beetroot supplementation on peak power output revealed a trivial, non-significant effect in favour of the beetroot juice intervention (Hedges’ g = 0.05, 95% CI −0.26 to 0.37, Z = 0.34, *p* = 0.74). For the subgroup analysis of chronic beetroot supplementation there was a trivial, non-significant effect in favour of beetroot juice (Hedges’ g = 0.11, 95% CI −0.20 to 0.41, Z = 0.68, *p* = 0.49). There were no differences in heterogeneity between subgroups (I^2^ = 0%, *p* = 0.81).

A combined analysis of the effect of chronic beetroot supplementation on mean and peak power analysis was conducted and is displayed in Figure 5. The analysis aims to examine the impact of chronic supplementation on the pooled power output measures and compare the effect on peak power to mean power output. There was an overall trivial (Hedges’ g = 0.08, 95% CI −0.15 to 0.31, Z = 0.67, *p* = 0.51), non-significant effect in favour of the beetroot intervention. Six analysed studies presented positive effects on power output after beetroot intervention, whilst two studies found no difference between beetroot juice and placebo [44,50], and three studies showed negative effects [8]. There were no differences in heterogeneity between peak power and mean power output (I^2^ = 0%, *p* = 0.78).

For the three studies which examined the mean difference in distance covered (metres) during the Yo-Yo IR1 protocol [7,22,55], there was a small and significant overall effect in favour of the beetroot juice intervention (Hedges’ g = 0.46, 95% CI: 0.14, 0.77, Z = 2.84, *p* = 0.004) (Figure 6). However, there was substantial heterogeneity among studies (I^2^ = 72%; *p* = 0.03).

## 4. Discussion

Given that nitrate supplementation is suggested to be most beneficial in conditions of hypoxia—either altitude or local hypoxia within the muscle—we anticipated that beetroot, as the most common form of dietary nitrate supplementation in exercise studies, would improve power output measures in HIIT and SIT exercise performance. However, from the 14 studies included that evaluated power output measures using repeated sprint protocols, there was no overall effect in favour of beetroot supplementation. This was true for both mean and peak power output (Figure 3; Figure 4, respectively), after acute or chronic supplementation with beetroot, or when power output measures were pooled after chronic supplementation with beetroot (Figure 5). The data from our analysis do not currently support acute or chronic beetroot juice ingestion for improving power output during HIIT/SIT exercise performance. Conversely, for three studies which examined distance covered after a Yo-Yo IR1 the analysis favoured the beetroot intervention [7,22,55] suggesting that in progressive sprints to exhaustion there may be some benefit.

### 4.1. Power Output

All the acute and chronic studies included in our analysis of power output employed maximal sprint exercise. However, the protocols varied in terms of the duration of sprint, the recovery time, and the number of repetitions applied. These variations contribute to difficulties in study comparison and interpretation compared with studies of endurance where time trials or time-to-exhaustion protocols provide standardised measures of performance. To minimise effects on interpretation, we analysed data on mean power and peak power (for Distance Covered in the Yo-Yo IR1 please see Section 4.2 below) separately. In addition, acute and chronic supplementation protocols were categorised into subgroups for analysis so that the ergogenic effect could be examined individually. The data from these pooled analyses were consistent in showing that beetroot had no influence on mean or peak power compared with placebo.

Six studies [34,35,38,52,53,54] applied an acute beetroot supplementation strategy before repeated sprints. The findings from our analysis here contrast with several studies in the literature [57,58,59] which have examined the effect of acute beetroot supplementation on isolated sprints (Wingate tests). In these studies of isolated single sprints, there was a consistent improvement in peak power output, along with a shorter time taken to attain to peak power (Appendix D) [57,58,59]. It has been suggested that the difference in observations between single and repeated sprints might be due to a predominant effect of nitrate on the initial force production of type II muscle fibres [3]. In line with this, a study in mice reported enhanced early phase force production in fast-twitch muscle after nitrate supplementation, potentially through improved calcium handling [60]. However, it has also been reported that improvements in force production in humans associated with contractile function after beetroot ingestion may be independent of proteins associated with calcium handling [3,61]. The mechanistic pathways behind this require further investigation. Collectively, what does appear clear from the available data is that there is no enduring effect of beetroot ingestion on power output in repeated maximum sprints, whether an acute or chronic supplementation strategy is used.

Surprisingly our data contrast with two studies in the literature where beetroot juice was not employed. A study by Porcelli and colleagues [62] found that 6 days of a high nitrate diet (8.2 mmol/day) from increased fruit and vegetable (not beetroot) consumption improved peak power output across five 6 s, all-out repeated sprints separated by 24 s of inactive recovery in comparison with a low nitrate diet (2.9 mmol/day) [62]. At the same time, plasma nitrate increased by 500% and nitrite by 50% in response to the high nitrate diet without any significant changes in the low nitrate diet. In the second study by Muggeridge and co-workers, nitrate-enriched peach flavoured gels (~8 mmol) were provided 2.5 h before nine SIT sessions (4–6 repeated, 15 s maximal sprints interspersed with 4 min recovery) conducted over 3 weeks in healthy, young males. Compared with a group provided low nitrate gels (~0.1 mmol), there was a tendency toward a greater maximum work rate and reduced fatigue index (change in mean power from the first to last sprint) in the nitrate-enriched supplemented group [63]. The reason for the difference in findings between these studies and our overall analysis is not clear, but future studies could examine sources of dietary nitrate as a possibility.

### 4.2. Distance Covered

In contrast to our analysis of power output, data from the pooled analysis showed a significant and large effect on the distance covered during the Yo-Yo IR1 test. It is important to highlight that this analysis was limited to three studies. However, all three studies exhibited a significant improvement in distance covered by 4.2 [22], 3.9 [7], and 3.4% [55], respectively. The progressive nature of the Yo-Yo IR1 test may explain the findings here in comparison with the more intense or all-out nature of the repeated sprints described in the previous section. Improvements in both muscle contraction (a reduction in the high-energy phosphate cost of force production) and mitochondrial oxidative phosphorylation (the ratio of adenosine triphosphate resynthesised to oxygen consumed) have been suggested as mechanisms via which beetroot supplementation improves exercise efficiency [7]. These mechanisms may have been of greater importance in a longer, progressive Yo-Yo IR1 protocol which may be a test more reflective of fatigue resistance than ability to recover peak power as discussed earlier (Section 4.1). Moreover, the consistency of the test protocol employed among the studies may have been another factor. One caveat to our findings here is that there was substantial heterogeneity among the studies included, limiting confidence in our effect size estimate, even though none of the studies was considered high risk or of concern in terms of study design.

### 4.3. Other Measures

Several other performance measures were reported in the studies included here which were not subjected to meta-analysis because of the inconsistency of the measures involved. These include 1 km kayaking time trial performance after sprint interval exercise [38] and time-to-exhaustion [36,50]. The study from Thompson and colleagues [50] showed that after 4 weeks of SIT with beetroot supplementation there were greater exercise capacity adaptations, as evidenced by improved time-to-exhaustion [50].

### 4.4. Exercise Level

It has previously been suggested that highly trained athletes may be less responsive to the ergogenic effect of nitrate than recreational athletes [64] because of higher endogenous NO bioavailability which renders the potential for additional NO production after dietary nitrate supplementation redundant. In addition, greater skeletal muscle capillarisation, increased content of skeletal muscle calcium ion handling proteins, and/or a lower proportion of type II muscle fibres in endurance-trained athletes may contribute to this observation [64]. It is uncertain whether such differences mitigate the effectiveness of nitrate supplementation during HIIT/SIT, particularly as well-trained athletes in high-intensity sports have a high proportion of type II muscle fibres [65]. Only one study in the present review examined the influence of training level on SIT. Jonvik and colleagues [33] found that plasma nitrate and nitrite concentrations increased to a similar extent with 6 days of beetroot supplementation in recreational cyclists, competitive national talent speed skaters and elite (Olympic-level) track cyclists but there was no improvement in peak or mean power output after 30 s repeated maximum sprints (Wingate tests) in any of the groups. Given that all three groups selected for the study were classified as sprint athletes this may have accounted for the similar response among groups. A second study included in this review included elite endurance athletes (maximum oxygen uptake >65 mL/kg/min)—without any comparator group—but again there was no effect of 3 days beetroot supplementation (5.2 mmol/day) on 15 sprints with 30 s passive recovery performed until exhaustion [36]. Thus, current studies do not suggest a differentiated response in power output measures of highly trained athletes undergoing HIIT/SIT after beetroot supplementation. However, as only a single study employed a direct comparison based on athletic level, more evidence is needed here. 

### 4.5. Supplementation Strategy

Nine studies included information on the increase in plasma nitrite after supplementation (Table 2). One acute study by Muggeridge and colleagues found a 32% increase in plasma nitrite after a 5 mmol beetroot supplement provided 3 h pre-exercise compared with a 0.2% increase after placebo [38]. The increase in nitrite was directly related to the total dosage of nitrate received with chronic beetroot supplementation. Although the nitrite concentration increased somewhat with the intake per day it was elevated significantly by the total dosage per trial [32,50]. This evidence provides support toward the hypothesis that excess nitrate and nitrite can be preserved and stored in the blood and tissues as a NO reservoir [3]. Future studies should examine the association between total beetroot dosage provided and nitrite concentration.

### 4.6. Source of Nitrate and Beetroot Juice

The source of beetroot juice provided in these studies were listed in Table 2. Fourteen of the seventeen studies included used beetroot shots produced by James White Drinks Ltd., under the brand name Beet It. The serving size of each shot provided was 70 mL, but the reported nitrate content varied among studies from 4.10 to 6.45 mmol. Manufacturer information on the available products (https://beet-it.com/product/beet-it-sport-shot/, accessed on 1 September 2021) lists different drink shots under the names of Beet It SPORT, NITRATE 400 (70 mL), and Beet It SPORT NITRATE 3000 super concentrate (250 mL for seven servings). A Beet It Shot contains 400 mg nitrate in a 70 mL serving size, equivalent to 6.45 mmol (nitrate molecular weight 62.0049 g/mol). A study by Gallardo and Coggan [66] sampled 24 different beetroot products from 21 different companies to analyse nitrate content. Three products from James White Drinks Ltd. were listed, which included Beet It SPORT Shot (70 mL, 6.45 mmol), Beet It Organic Shot (70 mL, 4.84 mmol), and Beet It Beet Juice (500 mL, 7.55 mmol) [66]. The variation of nitrate content reported in the included studies in the present analyses could be because of decimal rounding in reporting, variation between claimed and measured content, or that previous versions of Beet It were used. Gallardo and Coggan [66] suggested measuring the nitrate content of any beetroot supplements used in scientific research rather than depending on the manufacturer’s claim. This is an important consideration, not only for future studies in this area but also for coaches and athletes who may be attempting to optimise performance for competition with the use of beetroot supplements.

Variations in other ingredients which may have ergogenic properties are also an important consideration among studies. For example, the Beet It Shot formulated with lemon juice (2%) contains relatively high sodium (480 mg/100 mL) and protein (3.7 g/100 mL) content compared with single strength beetroot juice with an average 43.9 mg/100 mL of sodium content and 0.7 g/100 mL of protein [67,68]. Nutritional information on the various drinks used in studies in this review is tabulated and compared with single strength beetroot juice in Table 3 [67,68]. Two studies that did not use Beet It products [36,54], by sourcing the beetroot from Pajottenlander or developing it in-house, did not find an improvement in performance after ingesting the supplements. One study from Aucouturier and colleagues showed improvement after supplementation with beetroot source from Pajottenlander [42]. 

An important aspect of work neglected in most studies to date is that there was no detailed sensory or acceptance evaluation of the beetroot products used in supplementation. It has recently been suggested that taste is an important component in relation to the ergogenic potential of substances that improve athletic performance [69], and this should be examined even though a survey on consumer sensory evaluation showed that consumers did not have an explicit preference for the taste, smell, colour, and consistency of beetroot juice when compared between organic and conventional beetroot juice [70]. 

### 4.7. Limitations and Future Considerations

Although many studies have analysed the effect of nitrate on endurance exercise performance, the pool of 17 studies (14 focusing on the main outcome of power output) on HIIT or SIT we identified here is limited. As previously stated, although there was trivial statistical heterogeneity in our analysis related to power output, there was substantial variation in the competitive level of the participants, the exercise protocols employed, the beetroot dosage, the type of beetroot products, and the supplementation strategy and duration. These important variations in study design and methodology, in addition to the limited pool of studies, make the effects harder to tease out in a certain group of participants and potentially limit a firm conclusion of the effect of beetroot on HIIT performance. Another limitation is that this review was restricted to an analysis of beetroot on HIIT. Whilst this improves the consistency of the findings, as noted, at least one study employing a high nitrate diet from non-beetroot fruits and vegetables found a significant effect on power output with repeated sprints [62]. Thus, our analysis did not cover all studies including nitrate in the literature. However, given that to date beetroot remains the predominant source of nitrate supplementation for exercise studies, we believe that the findings reported here are valuable and probably reflective of the current evidence. A final limitation is that, as noted in Section 4.1, this review was confined to the examination of beetroot on power output in repeated sprint scenarios (HIIT and SIT). We did not systematically examine outcomes from studies where a single sprint was employed [71] and thus the data here should be carefully interpreted for particular sports or training regimens.

### 4.8. Practical Implications

A recent IOC consensus statement identifies dietary nitrate [5], which includes beetroot juice, as a nutritional ergogenic aid for higher intensity exercise performance (including intermittent team sport activities) [5,6,7,8,9,10]. Our data suggest that based on the currently available evidence, beetroot juice is not an efficacious ergogenic aid for coaches and athletes intending to employ it from the perspective of power output in sports/training sessions using repeated sprints. However, this does not mean that beetroot juice does not offer advantages in other aspects of use and the strategy of employment should be considered in relation to the overall demands of the sport. There may also be some advantages of using beetroot juice for athletes involved in sports where only a single sprint is necessary (see Section 4.1). 

Moreover, there may be some advantage in sports/activities where fatigue resistance is important. However, given the small number and heterogeneity of the studies where the Yo-Yo IR1 protocol was used, coaches and athletes should carefully monitor any use of beetroot juice for the performance of sports which are similar in nature.

Finally, coaches and athletes should carefully monitor the use of beetroot as a supplement during training. The possibility of gut discomfort may occur in some athletes, and as noted in Section 4.4, athletes at different levels of competition may not all achieve the same ergogenic effect with nitrate supplementation.

### 4.9. Conclusions

In conclusion, findings from the present meta-analysis suggest no significant improvement on peak and mean power output during repeated sprint training protocols after acute or chronic beetroot supplementation. Variation in the exercise protocols, nitrate dosage, type of beetroot products, supplementation strategy, and duration may restrict a firm conclusion of evidence for the effect of beetroot on power output during repeated sprints at present. Further research which optimises the supplementation strategy covering the dosage, timing, and type of beetroot products is required. The potential combined effect of other ingredients in the beetroot products should not be undermined. A chronic supplementation protocol with a higher beetroot dosage (>12.9 mmol/day for 6 days) is recommended, with the aim to better ensure an elevation in plasma nitrite, before future studies attempt to further examine if there is any effect of beetroot on HIIT and SIT.

## Figures and Tables

**Figure 1 nutrients-13-03674-f001:**
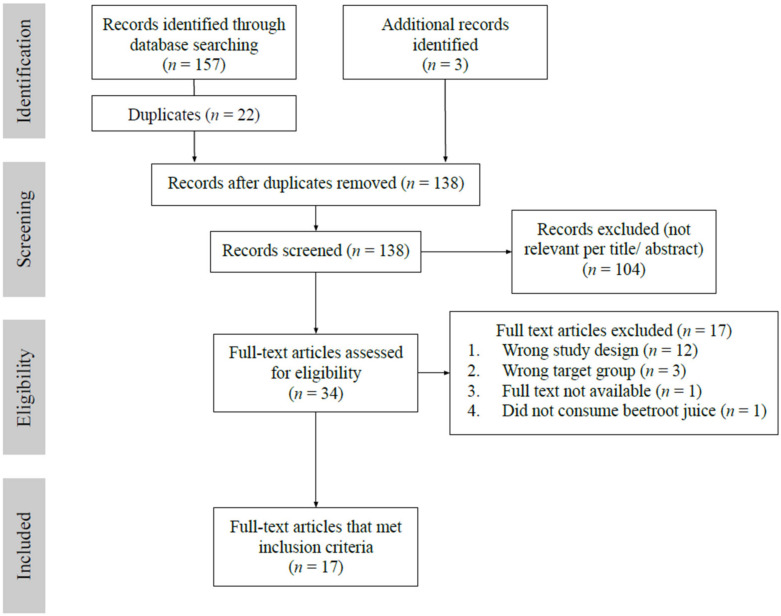
Selection process based on the Preferred Reporting Items for Systematic Reviews and Meta-Analysis (PRISMA) flowchart.

**Figure 2 nutrients-13-03674-f002:**
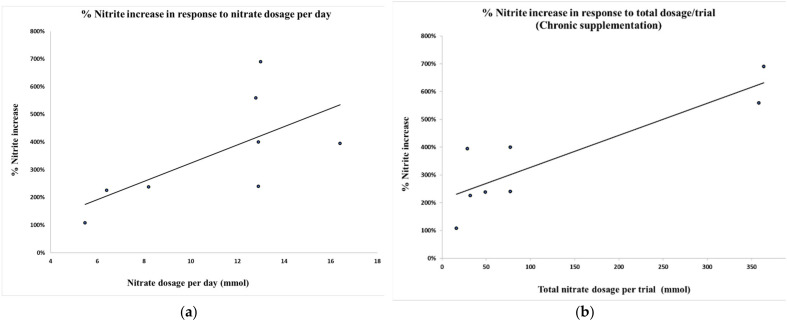
Percentage nitrite increase in response to (**a**) nitrate dosage per day (*r* = 0.655, *p* = 0.078, *N* = 8) and (**b**) total nitrate dosage per trial (*r* = 0.878, *p* = 0.004, *N* = 8).

**Figure 3 nutrients-13-03674-f003:**
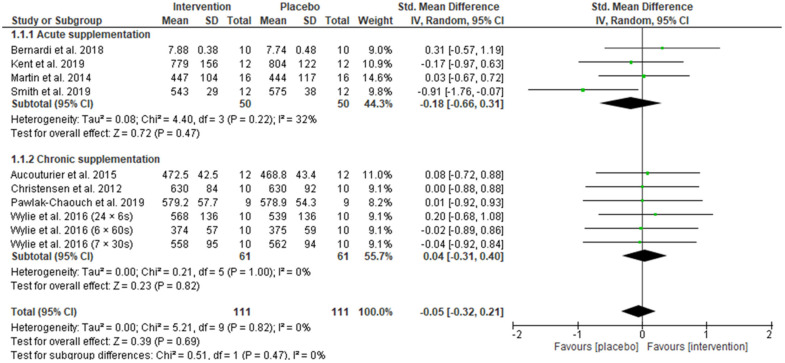
Forest plot comparing the effect of acute and chronic beetroot supplementation on mean power output.

**Figure 4 nutrients-13-03674-f004:**
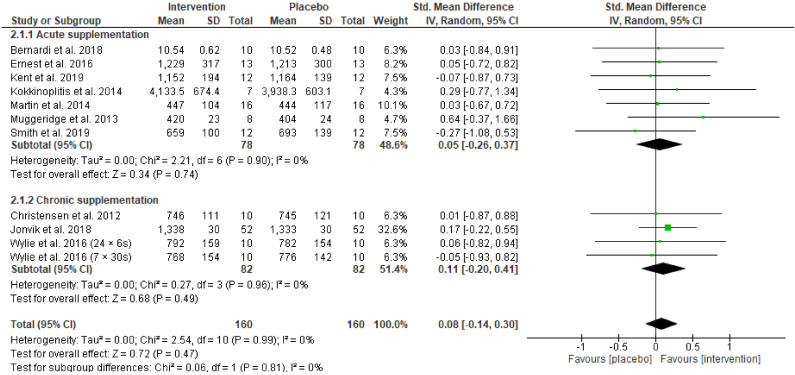
Forest plot comparing the effect of acute and chronic beetroot supplementation on peak power output.

**Figure 5 nutrients-13-03674-f005:**
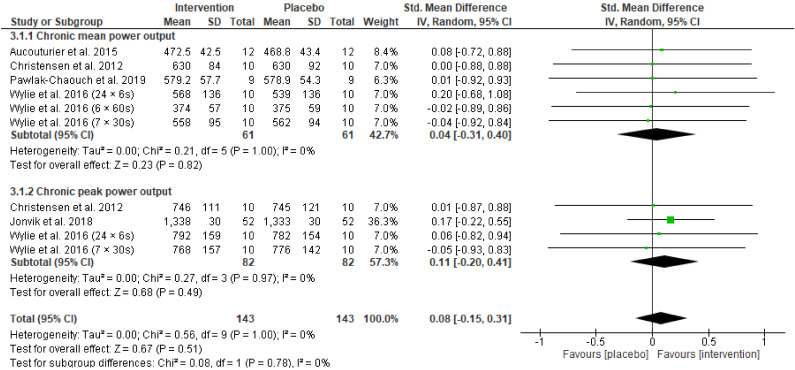
Forest plot comparing the effect of chronic beetroot supplementation on peak and mean power output.

**Figure 6 nutrients-13-03674-f006:**
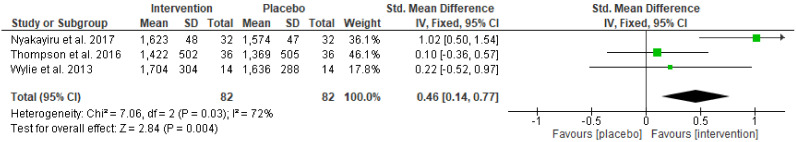
Forest plot comparing the effect of beetroot supplementation on the distance covered in Yo-Yo IR1 test.

**Table 1 nutrients-13-03674-t001:** Summary of the included studies assessing the effect of dietary beetroot supplementation on HIIT and SIT performance.

Study	Year	Sample Size (*n*)	Age (Years)	Exercise Level and Fitness, VO_2peak/max_ (mL/kg/min)	A/C	Beetroot Juice Supplementation	Study Design	Exercise Protocol	Primary Outcome
Bernardi et al. [53]	2018	10	24.9 ± 4.6	Well-trained competitive mixed martial arts athletes.	A	9.3 mmol (400 mL)	Randomised, crossover, double blind	20 all-out 6 s sprints, 24 s of recovery.	No significant improvement in peak and mean power. Peak power (BR: 10.54 ± 0.62 W/kg vs. PL: 10.52 ± 0.48 W/kg; *p* > 0.05). Mean power (BR: 7.88 ± 0.38 W/kg vs. PL: 7.74 ± 0.48 W/kg; *p* > 0.05).
Ernest et al. [43]	2016	13	25.9 ± 7.5	Competitively trained athletes.	A	2 × 11.2 mmol (2 × 70 mL)	Randomised, repeated measures, crossover, double blind	Maximal inertial-load cycling trials 4 × (3–4 s, 5 min passive rest) followed by maximal isokinetic cycling (30 s).	Peak power increased with BR (Pre: 1160 ± 301 W, post: 1229 ± 317 W) compared to PL (pre: 1191 ± 298 W, post: 1213 ± 300 W). Change in peak power post BR = 6.0 ± 2.6% compared to PL = 2.0 ± 3.8% (*p* = 0.014).
Kent et al. [35]	2019	12	22.3 ± 2.6	Team-sport athletes. VO_2 peak_ = 53.1 ± 8.7	A	12.9 mmol (2 × 70 mL)	Repeated measures, counter-balanced, double blind	4 sets of 9 × 4 s Repeat-Sprint Efforts, 16 s active recovery, and 6 s passive rest recovery per sprint. Sets were separated by 3 min of low-intensity cycling.	No significant difference in peak and mean power output. Peak power (BR: 1152 ± 194 W vs. PL: 1164 ± 139 W), mean power (BR: 779 ± 156 W vs. PL: 804 ± 122 W).
Kokkinoplitis et al. [51]	2014	7	25.2 ± 3.3	Healthy men. Exercise level not specified.	A	6.45 mmol (70 mL)	Randomised, crossover, double blind	Repeated high intensity sprints (5 × 6 s) on a non-motorised treadmill with 30 s of recovery.	No significant difference in mean peak power and strength. Mean peak power (BR: 4133.5 ± 674.4 W vs. PL: 3838.3 ± 603.1 W; *p* = 0.79).
Martin et al. [52]	2014	16	M22.3 ± 2.1, F20.7 ± 1.3	Team-sport athletes. V0_2max_ = 57.4 ± 8.5(M); V0_2max_ = 47.2 ± 8.5(F)	A	4.84 mmol (70 mL)	Randomised, crossover, double blind	8 s sprints with 30 s recovery on a cycle ergometer to exhaustion.	No significant difference in mean power output and mean peak power. Mean power (BR: 447 ± 104 W vs. PL: 444 ± 117 W; *p* = 0.797), mean peak power (BR: 447 ± 104 W vs. PL: 444 ± 117 W; *p* = 0.196).
Muggeridge et al. [38]	2013	8	31.0 ± 15.0	Trained. VO_2peak_ = 49.0± 6.1	A	5 mmol (70 mL)	Randomised, crossover	15 min steady-state paddling followed by five 10 s maximal effort SIT, 50 s active recovery. After 5 min rest, complete 1 km TT.	No difference in either peak power in the sprints or TT performance between conditions. Peak power (BR: 420 ± 23 W vs. PL: 404 ± 24 W; *p* = 0.59).
Smith et al. [34]	2019	12	22.0 ± 4.0	Recreationally trained.	A	6.2 mmol (70 mL)	Randomised, counter- balanced, crossover, double blind	20 maximal 6 s sprints, 114 s of active recovery.	No significant improvement in peak and mean power. Peak power (BR: 659 ± 100 W vs. PL: 693 ± 139 W; *p* = 0.056), mean power (BR: 543 ± 29 W vs. PL: 575 ± 38 W; *p* = 0.081).
Aucouturier et al. [42]	2015	12	22.8 ± 3.1	Recreation team sport players. VO_2peak_ = 46.6 ± 3.4	C	5.48 mmol/day for 3 days (500 mL)	Randomised, crossover, single blind	Cycle ergometer: number of sets until exhaustion (15 s at 170% maximal aerobic power, 30 s passive recovery periods).	Number of repetitions before exhaustion was significantly higher with BR compared PL (26.1 ± 10.7 vs. 21.8 ± 8.0, *p* < 0.05). Mean power output (BR: 472.5 ± 42.5 W vs. PL: 468.8 ± 43.4 W).
Christensen et al. [44]	2013	10	29.0 ± 4.0	Highly trained cyclists. VO_2max_ = 72.1 ± 4.5	C	8.06 mmol/day for 6 days	Randomised, crossover, single blind	VO2 kinetics (3 × 6 min at 298 ± 28 W), endurance (120 min preload followed by a 400 kcal time trial). Repeated sprint test (cycle ergometer): 6 × 20 s sprints, 100 s active recovery.	No significant difference in VO_2_ kinetics, exercise economy, time trial, peak, and mean power. Average peak power (BR: 746 ± 111 W vs. PL: 745 ± 121 W), mean power (BR: 630 ± 84 W vs. PL: 630 ± 92 W).
Jonvik et al. [33]	2018	52	27.0 ± 6.0	Recreational cyclists (*n* = 20), national talent speed-skaters (*n* = 22) and Olympic-level track cyclists (*n* = 10)	C	12.9 mmol/day (140 mL) for 6 days	Randomised, crossover, double blind	3 30 s Wingate tests (4 min active recovery).	No significant difference in peak and mean power output. Time to peak power improved by 2.8%. Peak power (BR: 1338 ± 30 W vs. PL: 1333 ± 30 W; *p* = 0.62).
Nyakayiru et al. [54]	2017	32	23.0 ± 1.0	Trained soccer players	C	12.9 mmol/day (2 × 70 mL) for 6 days	Randomised, placebo-controlled, crossover, double blind	Yo-Yo IR1.	Performance (distance covered) improved by 3.4 ± 1.3%. Distance (BR: 1623 ± 48 m vs. PL: 1574 ± 47 m; *p* = 0.027).
Pawlak-Chaouch et al. [36]	2019	9	21.7 ± 3.7	Elite. VO_2max_ > 65	C	5.2 mmol/day (500 mL) for 3 days	Randomised, placebo-controlled, crossover, single blind	SIE test until exhaustion. 15 s cycling at 170% of the maximal aerobic power, 30 s passive recovery.	No significant difference in mean power, total work completed and total repetitions. Mean power (BR: 579.2 ± 57.7 W vs. PL: 578.9 ± 54.3W).
Thompson et al. [7]	2016	36	24.0 ± 4.0	Competitive team sport players	C	6.4 mmol/day (70 mL) for 5 days	Randomised, crossover, double blind	Maximal 20 m sprints followed by the Yo-Yo IR1; 10 s active recovery, 5 min passive recovery.	1.2% improvement in 20 m sprint; 3.9% improvement in distance covered. Distance (BR: 1422 ± 502 vs. PL: 1369 ± 505 m; *p* < 0.05).
Thompson et al. [32]	2017	36	M27.0 ± 8.0, F23.0 ± 4.0	Recreationally active. VO_2peak_ = 50.4 ± 11.4(M); VO_2peak_ = 39.8 ± 5.8(F)	C	13 mmol/day (2 × 70 mL) for 28 days	Randomised to matched independent groups, double blind	SIT with supplementation for 28 days or without training intervention. Two severe-intensity step tests, 3 min and 20 min passive recovery repeat until task failure. Training session, Wingate all out, 30 s (4 and 5 times), rest 4 min.	No performance improvement in TTE trial. Peak WR (BR: 321 ± 91 W vs. PL: 318 ± 73 W; *p* > 0.05).
Thompson et al. [50]	2018	30	M25.0 ± 6.0, F22.0 ± 3.0	Recreationally active. VO_2peak_ = 46.6 ± 7.5 (M). VO_2peak_ = 39.9 ± 3.9 (F).	C	12.8 mmol/day (2 × 70 mL) for 28 days	Randomised to matched independent groups, double blind	Two bouts of severe-intensity cycling, the first for 3 min and the second until task failure; 4-wk supervised SIT program (Wingate 30 s, 4 min rest).	No significant improvement in peak work rate. TTE increased in SIT BR (71%), SIT (47%) and SIT KNO_3_ (42%).
Wylie et al. [22]	2013	14	22.0 ± 2.0	Recreational team-sport players. VO_2max_ = 52 ± 7	C	7 × 70 mL (4.1 mmol). Around 29 mmol in 36 h	Randomised, crossover, double blind	Yo-Yo IR1.	Distance covered was 4.2% greater with BR compared to PL. Distance (BR: 1704 ± 304 m vs. PL: 1636 ± 288 m; *p* < 0.05).
Wylie et al. [8]	2016	10	21.0 ± 1.0	Team-sport players. VO_2peak_ = 58 ± 8	C	8.2 mmol/day (2 × 70 mL) for 5 days. Test day 2 × 70 mL, and post-test 70 mL	Randomised, crossover, double blind	Twenty-four 6 s all-out sprints, 24 s of recovery; seven 30 s all-out sprints, 240 s of recovery, and six 60 s self-paced maximal efforts, 60 s of recovery; on days 3, 4, and 5 of supplementation, respectively.	Mean power output was significantly greater (5.4%) with BR relative to PL in the 24×6 s protocol only. 24 × 6 s protocol mean power (BR: 568 ± 136 W vs. PL: 539 ± 136 W; *p* < 0.05), peak power (BR: 792 ± 159 W vs. PL: 782 ± 154 W; *p* > 0.05); 7×30 s protocol mean power (BR: 558 ± 95 vs. PL: 562 ± 94 W; *p* > 0.05), peak power (BR: 768 ± 157 vs. PL: 776 ± 142 W; *p* > 0.05); 6×60 s protocol mean power (BR: 374 ± 57 W vs. PL: 375 ± 59 W; *p* > 0.05).

A: acute; C: chronic; s: seconds; min: minutes; SIE/SIT: sprint interval exercise; m: meter; M: males; F: females; TTE: time to exhaustion; W: Watt; WR: work rate; TT: Time trial; BR: beetroot; PL: placebo; KNO_3_: Potassium nitrate; Yo-Yo IR1: Yo-Yo Intermittent Recovery Test Level 1. All data are mean ± standard deviation.

**Table 2 nutrients-13-03674-t002:** Summary of supplementation strategy with sources of beetroot products.

Study	Year	Exercise Level	A/C	Suppl.	Total NO_3_ Loaded (mmol)	Last Dose (h)	Source of Beetroot	% NO_x_ Increase	Erg *
Bernardi et al. [53]	2018	Trained	A	9.3 mmol (400 mL)	9.3	2	Produced in house		No
Ernest et al. [43]	2016	Trained	A	2 × 11.2 mmol (2 × 70 mL)	22.4	2.5	Beet it, James White Drinks Ltd., Ipswich, UK		Yes
Kent et al. [35]	2019	Trained	A	12.9 mmol (2 × 70 mL)	12.9	2	Beet it, James White Drinks Ltd., Ipswich, UK		No
Kokkinoplitisk et al. [51]	2014	Recreationally active	A	6.45 mmol (70 mL)	6.45	3	Beet It, James White Drinks Ltd., Ipswich, UK		No
Martin et al. [52]	2014	Trained	A	4.84 mmol (70 mL)	4.84	2	Beet It, James White Drinks Ltd., Ipswich, UK	1600% increase in ${\mathrm{NO}}_{3}^{-}$	No
Muggeridge et al. [38]	2013	Trained	A	5 mmol (70 mL)	5	3	Beet IT organic shot, James White Drinks Ltd., Ipswich, UK	360% increase in ${\mathrm{NO}}_{3}^{-}$. 32% increase in ${\mathrm{NO}}_{2}^{-}$	No
Smith et al. [34]	2019	Recreationally active	A	6.2 mmol (70 mL)	6.2	3	Beet-It-Pro Elite Shot, James White Drinks Ltd., Ipswich, UK		No
Aucouturier et al. [42]	2015	Recreationally active	C	5.48 mmol/day for 3 days (500 mL)	16.44	3	Pajottenlander TM, Belgium	970% increase in ${\mathrm{NO}}_{3}^{-}$. 108% increase in ${\mathrm{NO}}_{2}^{-}$	Yes
Christensen et al. [44]	2013	Trained	C	8.06 mmol/day for 6 days	48.36	3	Beet it, James White Drinks Ltd., Ipswich, UK	298% increase in plasma NOx (nitrate + nitrite).	No
Jonvik et al. [33]	2018	Recreationally trained and elite	C	12.9 mmol/day (140 mL) for 6 days	77.4	3	Beet it, James White Drinks Ltd., Ipswich, UK	1600% increase in ${\mathrm{NO}}_{3}^{-}$. 400% increase in ${\mathrm{NO}}_{2}^{-}$	No
Nyakayiru et al. [54]	2017	Trained	C	12.9 mmol/day (2 × 70 mL) for 6 days	77.4	2.5	Beet It, James White Drinks Ltd., Ipswich, UK	1100% increase ${\mathrm{NO}}_{3}^{-}$. 240% increase in ${\mathrm{NO}}_{2}^{-}$. 669% increase in salivary ${\mathrm{NO}}_{2}^{-}$	Yes
Pawlak-Chaouch et al. [36]	2019	Elite	C	5.2 mmol/day (500 mL) for 3 days	15.6	2	Pajottenlander TM, Belgium.	325% increase in NOx	No
Thompson et al. [7]	2016	Trained	C	6.4 mmol/day (70 mL) for 5 days	32	2.5	Beet it, James White Drinks Ltd., Ipswich, UK	659% increase in ${\mathrm{NO}}_{3}^{-}$. 226% increase in ${\mathrm{NO}}_{2}^{-}$	Yes
Thompson et al. [32]	2017	Recreationally active	C	13 mmol/day (2 × 70 mL) for 28 days	364	2.5	Beet It, James White Drinks Ltd., Ipswich, UK	960–1050% increase in ${\mathrm{NO}}_{3}^{-}$_3_. 690–715% increase in ${\mathrm{NO}}_{2}^{-}$	No
Thompson et al. [50]	2018	Recreationally active	C	12.8 mmol/day (2 × 70 mL) for 28 days	358.4	2.5	Beet it, James White Drinks Ltd., Ipswich, UK	559% increase in ${\mathrm{NO}}_{2}^{-}$	No
Wylie et al. [22]	2013	Recreationally active	C	7 × 70 mL (4.1 mmol). Around 29 mmol in 36 h	29	2.5 and 1.5	Beet it, James White Drinks Ltd., Ipswich, UK	2972% increase in ${\mathrm{NO}}_{3}^{-}$. 395% increase in ${\mathrm{NO}}_{2}^{-}$	Yes
Wylie et al. [8]	2016	Recreationally active	C	8.2 mmol/day (2 × 70 mL) for 5 days. Test day 2 × 70 mL, and post-test 70 mL	49.2	2.5 and post test	Beet It, James White Drinks Ltd., Ipswich, UK	238% increase in ${\mathrm{NO}}_{2}^{-}$	Yes

A: acute; C: chronic; s: seconds; min: minutes; BR: beetroot; PL: placebo; Suppl.: Supplementation; Erg: Ergogenic; hrs: hours. * Statistically significant change in power output/performance.

**Table 3 nutrients-13-03674-t003:** Nutritional information panel for beetroot products and beetroot juice. The values are presented in g/100 mL or mg/100 mL.

Product	Beet It Sport Nitrate 400 Shot ^a^	Beet It Organic Beetroot Juice ^b^	Pajottenlander Red Beetroot Juice ^c^	Beetroot Juice
Energy	88 kcal/373 kJ	32 kcal/142 kJ	43 kcal/181 KJ	95 kcal/399 kJ ^1^
Protein	3.7 g	0.8 g	1.0 g	0.7 g ^1^
Total Fat	0.0 g	0.0 g	<0.5 g	0.2 g ^1^
Carbohydrate	18.0 g	7.6 g	8.5 g	22.6 g ^1^
Total Sugars	17.0 g	6.8 g	7.0 g	12.1 g ^1^
Sodium	480 mg	80 mg	50 mg	43.9 mg ^2^
Nitrate	571 mg (400 mg per 70 mL)	80 mg	Not declared	99.2 mg ^1^
Ingredients	Concentrated beetroot juice (98%), lemon juice (2%); made from concentrates.	Organic redbeet juice (90%), organic apple juice (10%); not from concentrate.	100% lactofermented red beetroot juice	Beetroot Juice

Source: ^a^
https://www.jameswhite.co.uk/product/beet-it-sport-shot/, accessed on 1 September 2021; ^b^
https://www.jameswhite.co.uk/product/beet-it-tetra-pack-1l/, accessed on 1 September 2021; ^c^
https://www.pajottenlander.be/?en/products/vegetable%2Djuice/308/red%2Dbeetroot%2Djuice/, accessed on 1 September 2021; ^1^ Nutritional, Bioactive and Physicochemical Characteristics of Different Beetroot Formulations, Baião, D. et al., 2017 [67]; ^2^ Compositional characteristics of commercial beetroot products and beetroot juice prepared from seven beetroot varieties grown in Upper Austria, Wruss J. et al., 2015 [68].

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
