# Peer review of "The Effect of Beetroot Ingestion on High-Intensity Interval Training: A Systematic Review and Meta-Analysis"

_nutrients, 2021, doi:10.3390/nu13113674_

Round 1

Reviewer 1 Report

Overall a good summary of the literature. The varied methods, dosage, mode, and subjects make it difficult to draw firm conclusions. The methods of this review were sound and captured the most relevant studies on this topic. 

The last sentence of the conclusion doesn't match the results and discussion. From the review, it doesn't appear there is enough evidence to support chronic supplementation to improve SIT or HIIT.

See additional comments embedded in the manuscript.

Author Response

Thank you for your comments. Please see attached file for the responses.

Reviewer 2 Report

The paper is interesting and within the scope of the journal. However, I have some concerns to improve the paper that should be addressed by the authors:

  • please highlight the main aim of the study, rewriting the introduction section.
  • please highlight clearly the main limitations of the paper.
  • please address practical implications to athletes and coaches with more practical applications.

Author Response

Thank you for your comments. Please see attached file for responses.
